# Integrative Multi-Omics Analysis of the Rumen in *Tan* Sheep with Contrasting Average Daily Gain

**DOI:** 10.3390/microorganisms13122882

**Published:** 2025-12-18

**Authors:** Hao Zheng, Xiaohong Han, Wenjuan Shen, Xinrui Zhang, An Shi, Tonggao Liu, Chong Yang, Jinzhong Tao

**Affiliations:** 1College of Animal Science and Technology, Ningxia University, Yinchuan 750021, China; 13213619969@163.com (H.Z.); hanxh1254@126.com (X.H.); shenwenjuan0518@163.com (W.S.); iszhangxinr@163.com (X.Z.); shian_1988@outlook.com (A.S.); 2Animal Husbandry Workstation of Ningxia, Yinchuan 750021, China; nxxmztykl@163.com (T.L.); xmjych7203@163.com (C.Y.)

**Keywords:** ADG, plasma, 16S rRNA, metabolomics

## Abstract

Understanding the drivers of average daily gain (ADG) is key to enhancing the productivity of *Tan* sheep. This study employed an integrated multi-omics approach to compare rumen microbial communities (16S rRNA sequencing) and metabolomic profiles between *Tan* sheep with high (HADG) and low (LADG) ADG. The novelty of this work lies in the systems-level identification of functional linkages between specific rumen bacteria and metabolites that underlie divergent growth phenotypes. The results revealed no significant difference in initial body weight between the two groups (*p* > 0.05). However, the HADG group showed significantly higher final body weight (*p* < 0.05), markedly greater ADG and Average Daily Dry Matter Intake (ADFI) (*p* < 0.01), and a substantially lower FCR (*p* < 0.01). Plasma Total Antioxidant Capacity (T-AOC) and Superoxide Dismutase (SOD) levels were significantly elevated in the HADG group (*p* < 0.05), while Malondialdehyde (MDA) concentration was significantly reduced (*p* < 0.05). In contrast, plasma Globulin (GLB), Glucose (GLU), and Triglycerides (TG) concentrations were significantly lower in HADG sheep (*p* < 0.05). Rumen metabolomics identified 265 differentially abundant metabolites between groups, with 64 down-regulated and 201 up-regulated in LADG compared to HADG sheep. These metabolites were significantly enriched in tyrosine metabolism, β-alanine metabolism, and thiamine metabolism pathways. Receiver Operating Characteristic (ROC) curve analysis identified 15 key differential metabolites, including succinic acid, 2-hydroxyglutarate, and pyridoxal phosphate. 16S rRNA sequencing indicated significant differences in microbial genera such as UCG-002, Blautia, norank_f__Bacteroidales_UCG-001, and norank_f__norank_o__Rhodospirillales. Correlation analysis revealed that UCG-002 and norank_f__Bacteroidales_UCG-001 were highly negatively correlated with succinic acid (*p* < 0.01), and significantly negatively correlated with 1-aminocyclopropanecarboxylic acid, pyridoxal phosphate, and 2-hydroxyglutarate (*p* < 0.05). Conversely, beta-alanine, ureidoacrylic acid, L-proline, and 2′-deoxyguanosine showed a highly significant positive correlation with norank_f__Bacteroidales_UCG-001 (*p* < 0.01), and a significant positive correlation with UCG-002 (*p* < 0.05). These findings elucidate the molecular mechanisms behind growth differences in *Tan* sheep and provide actionable insights for developing targeted nutritional strategies.

## 1. Introduction

*Tan* sheep are an important local breed in Northwest China, valued for their meat quality and economic contribution. In modern, intensive farming, optimizing feed efficiency is critical to profitability and sustainability [1]. For meat sheep, Average Daily Gain (ADG) is a central economic trait directly impacting production costs and returns [2].

Animal growth and slaughter performance are intimately linked to their blood biochemical indices, which provide crucial insights into nutritional metabolism, energy homeostasis, immune function, and organ health [3,4]. The ruminal microbial community in ruminants plays a fundamental role in feed digestion, nutrient assimilation, energy harvest, immune modulation, and host metabolic processes [5,6,7]. The Shannon diversity index for the gastrointestinal microbiota of *Tibetan* sheep was reported to be between 7.5–8.5 [8]. Sun et al. [9] discovered that infection with Escherichia coli F17 induced diarrhea in lambs, a reduced Bacteroidetes/Firmicutes ratio, and diminished microbial diversity. Likewise, Artegoitia et al. [10] in their investigation of rumen fluid from bulls with high and low ADG, reported that linoleic acid and α-linolenic acid (impact-value 1.0 and 0.75, respectively; *p* < 0.05) metabolism pathways were the most significantly impacted. A panel comprising pentadecanoic acid, palmitic acid, linoleic acid, and α-linolenic acid accurately differentiated high-ADG from low-ADG cattle.

Specifically, it is unclear how divergent ADG in *Tan* sheep is associated with distinct profiles of systemic physiological and metabolic status, and the composition and function of the ruminal microbial ecosystem. Resolving this knowledge gap is essential for moving beyond empirical breeding and feeding practices towards mechanism-based strategies for enhancing productivity. To address this, we hypothesized that high-ADG *Tan* sheep possess a more efficient physiological and ruminal metabolic phenotype compared to their low-ADG counterparts. To test this, we conducted an integrated multi-omics study comparing sheep with divergent ADG. We systematically analyzed growth performance, blood biochemistry, rumen metabolome, and microbial community structure. The novelty of this work lies in providing the first systems-level insight into the interconnected rumen microbiome-metabolome dynamics underlying ADG variation in *Tan* sheep, identifying specific microbial taxa and functional metabolites associated with superior growth performance.

## 2. Materials and Methods

### 2.1. Experimental Animals and Feeding Management

This trial protocol was reviewed and approved by the Animal Ethics Committee of Ningxia University (Approval ID: NXU-2024-143). The study was carried out on a *Tan* sheep farm in Wuzhong City (103° E, 35° N), Ningxia Hui Autonomous Region, situated at an altitude of 1600 m. 156 three-month-old male *Tan* sheep lambs with comparable birth dates, in good health, and with similar body weights (31.69 ± 3.72 kg) were selected for standardized rearing and management. The lambs were collectively housed and managed under uniform conditions. They received feed twice per day at 07:00 and 18:00 h (with feeders being emptied prior to each meal), and each feeding event was restricted to a maximum duration of 30 min. All study lambs were fed using a headlock feeding system(Ningxia Win Win Machinery Manufacturing Co., Ltd., Yinchuan, China). Following each meal, the headlocks were released to allow free movement and ad libitum access to water. Daily feed intake was measured and recorded. The entire experimental duration was 75 days, which included a 15-day adaptation phase, a 10-day pre-experimental phase, and a 50-day main experimental phase. The average daily dry matter intake (ADMI) and ADG for each lamb were calculated based on body weights recorded at 10-day intervals throughout the main experimental period, combined with the daily amounts of feed offered and left. Lambs exhibiting an ADG greater than the overall mean plus 0.5 standard deviations (Mean + 0.5 SD) were assigned to the High-ADG group (HADG), while those with an ADG less than the Mean minus 0.5 SD were assigned to the low-ADG group (LADG). Five individuals were randomly selected from each of the HADG and LADG groups, yielding a total subset of 10 lambs for subsequent detailed analysis. The ingredient composition and nutritional profile of the basal diet are presented in Table 1.

### 2.2. Experimental Sample Collection

At the conclusion of the formal trial period, the experimental sheep were fasted for 24 h. Prior to slaughter, approximately 10 mL of blood was collected from the jugular vein. The blood samples were centrifuged at 3000 rpm for 10 min to separate the plasma, which was then stored at −80 °C for subsequent analysis. Immediately after slaughter, rumen fluid and rumen content were collected from the *Tan* sheep, aliquoted into 5 mL cryovials, and stored at −80 °C for further use. Blood biochemical parameters, including Total Protein (TP), Albumin (ALB), Globulin (GLB), Total Cholesterol (TC), Triglycerides (TG), and Glucose (GLU), were analyzed using an automated biochemistry analyzer (Antu Biotechnology Co., Ltd., Zhengzhou, China). Immune and antioxidant markers were quantified using ELISA kits (Youxuan Biotechnology Co., Ltd., Shanghai, China). These included Immunoglobulins (IgA, YX-090701Y, IgG, YX-090707Y, IgM, YX-090713Y), Interleukins (IL-2, YX-091202H, IL-6, YX-091206H, IL-10, YX-091210H), Superoxide Dismutase (SOD, YX-W-A500), Malondialdehyde (MDA, YX-E20347), Total Antioxidant Capacity (T-AOC, YX-E10804), and Glutathione Peroxidase (GSH-Px, YX-E10801). (The experimental instrument used was a microplate reader: Infinite F50 (Tecan, Waltham, MA, USA).

### 2.3. Measurement of Growth Performance and Slaughter Performance

On day 50 of the formal feeding period, sheep in each group were fasted for 24 h and deprived of water for 2 h before being weighed and slaughtered. Parameters such as live weight before slaughter, carcass weight, weights of various internal organs, eye muscle area, and GR value were measured. The calculation formulas for growth performance and slaughter performance indicators are as follows:

Feed Conversion Ratio (FCR) = Average Daily Feed Intake/Average Daily Gain;

Live weight before slaughter = The weight recorded after a 24-h fast and 2-h water deprivation prior to slaughter;

Carcass weight = The weight of the entire body (including kidneys and kidney fat) after removing the skin, head, viscera, parts below the carpal joint of the forelimbs and the hock joint of the hindlimbs, and letting it rest for 30 min;

Dressing percentage = Carcass weight/Live weight before slaughter × 100%;

Loin eye area = The cross-sectional area of the longissimus dorsi muscle between the 12th and 13th ribs. It was calculated by tracing on sulfuric acid paper using the formula: Loin eye area = Loin eye height (cm) × Loin eye width (cm) × 0.7;

GR value = The thickness of the tissue at a point 11 cm from the midline of the spine between the 12th and 13th ribs.

The above determination shall be carried out according to the <<Technical specification for determination of production performance of meat sheep>> (T/CAAA 080-2022 [12]).

### 2.4. 16S rRNA Sequence Analysis of Rumen, Cecum and Rectum Microorganisms

DNA extraction of rumen microbiota, PCR amplification, Q-PCR quantification, Illumina library construction, Illumina sequencing, and data analysis were outsourced to Shanghai Majorbio Bio-pharm Technology Co., Ltd. (Shanghai, China, https://www.majorbio.com/ (accessed on 20 September 2025)). Specific details are as follows: DNA extraction: Genomic DNA was extracted using the SDS method; PCR amplification: The 16S rRNA gene was amplified using barcoded specific primers (341F-806R, sequences: 5′-CCTACGGGNGGCWGCAG-3′ and 5′-GGACTACHVGGGTWTCTAAT-3′, targeting the V3-V4 region) (Wuhan JinKairui Biological Technology Co., Ltd., Wuhan, China). The products were detected by 2% agarose gel electrophoresis and excised for recovery (using AxyPrep DNA Gel Extraction Kit, eluted with Tris-HCl, Meiji Biotechnology Co., Ltd., Shanghai, China); Q-PCR quantification (Meiji Biotechnology Co., Ltd., Shanghai, China): PCR products were quantified using the QuantiFluor™-ST system (Meiji Biotechnology Co., Ltd., Shanghai, China), and samples were mixed in proportion; Illumina library construction: Illumina official adapter (Meiji Biotechnology Co., Ltd., Shanghai, China) sequences were added to the ends of the target regions via PCR. After a second PCR amplification, the products were excised and recovered. Sodium hydroxide was used to denature and generate single-stranded DNA fragments.

Illumina sequencing: One end of the DNA fragment is complementary to the primer bases and fixed on the chip; using the DNA fragment as a template, bridge PCR amplification forms DNA clusters, and the DNA amplicons are linearized into single strands. Sequencing-by-synthesis (using fluorescently labeled dNTPs) is performed, and fluorescence signals are captured to determine the sequence (Illumina, San Diego, CA, USA).

Data analysis: Sequence processing and bioinformatics analysis were performed using QIIME 2 and Easy Amplicon v1.0.

### 2.5. Untargeted Metabolomic Profiling of Rumen Fluid

Rumen fluid metabolomic analysis was conducted using LC-MS. Sample preparation, LC-MS analysis, mass spectrometric parameter settings, and initial raw data processing were performed by Wuhan Metware Biotechnology Co., Ltd. (Wuhan, China). Data were subjected to pattern recognition using SIMCA-P software (Version 14.1, Umetrics, Umea, Sweden) with Pareto-scaling preprocessing. Multivariate statistical analysis, including Principal Component Analysis (PCA) and Orthogonal Partial Least Squares-Discriminant Analysis (OPLS-DA), was used to build expression models. Model data underwent permutation testing, and the Variable Importance in Projection (VIP) was obtained from the OPLS-DA model. Further univariate statistical analysis, including Fold Change (FC), was performed. Metabolites meeting the criteria of FC ≥ 1.5 or FC ≤ 0.66, VIP > 1, and *p* ≤ 0.05 were identified as differential metabolites. Subsequent analysis was conducted through the Metware online analysis cloud platform (https://cloud.metware.cn/, accessed on 20 September 2025).

### 2.6. Data Statistics and Analysis

Experimental data were collated using Excel 2021. Statistical analyses of the processed data were performed by using SIMCA software (Version 14.1, Umetrics, Umea, Sweden), including PCA and OPLS-DA. Receiver Operating Characteristic (ROC) curves were generated using Origin 2024. Differences were considered statistically significant at *p* < 0.05, highly significant at *p* < 0.01, and *p* > 0.05 indicated no significant difference. Correlation analysis among significantly different bacterial genera, significantly different metabolites, and average daily gain was performed using the Wekemo Bioincloud platform (https://www.bioincloud.tech, accessed on 20 September 2025). Heatmaps were generated to visualize Pearson correlation coefficients, with significance levels indicated by asterisks (* *p* < 0.05, ** *p* < 0.01) [13].

## 3. Results

### 3.1. Analysis of Growth/Slaughter Performance in High- and Low-ADG Tan Sheep

In this experiment, 156 male *Tan* lambs with similar body weights were selected and fed using the neck clamp method for 75 days, which included a transition period of 15 days, a pre-trial period of 10 days, and a formal trial period of 50 days. After the trial period, lambs with ADG greater than the mean (Mean) + 0.5 standard deviation (SD) were classified into the high-ADG group (HADG), and those with ADG less than Mean − 0.5 SD were classified into the low-ADG group (LADG). As shown in Table 2, there was no significant difference in initial body weight between the HADG and LADG groups of *Tan* sheep (*p* > 0.05). The final body weight of *Tan* sheep in the HADG group was significantly higher than that in the LADG group (*p* < 0.05). However, as shown in Table 3, the proportions of both head weight and hoof weight to pre-slaughter live weight were significantly lower than those in the LADG group (*p* < 0.05).The kidney weight of *Tan* sheep in the HADG group was significantly higher than that in the LADG group (*p* > 0.05) (see Appendix A). It is highly meaningful to proceed with subsequent experiments using the selected HADG and LADG *Tan* sheep.

**Table 2 microorganisms-13-02882-t002:** Analysis of differences in growth performance of HADG and LADG *Tan* sheep.

Item	HADG	LADG	*p*-Value
Initial weight/kg	31.48 ± 3.88	31.20 ± 3.19	0.904
Final weight/kg	49.24 ± 4.06 ^a^	42.08 ± 3.56 ^b^	0.018
ADG/g	362.46 ± 8.85 ^A^	222.06 ± 14.9 ^B^	<0.001
FCR	4.94 ± 0.33 ^B^	6.63 ± 0.47 ^A^	<0.001
ADFI/kg/day	1.61 ± 0.11 ^A^	1.33 ± 0.14 ^B^	0.008

Note: Same row without letters indicates insignificant differences (*p* > 0.05), different lowercase letters indicate significant differences (*p* < 0.05), different capital letters indicate highly significant differences (*p* < 0.01).

**Table 3 microorganisms-13-02882-t003:** Analysis of differences in slaughter performance of high- and low-ADG *Tan* sheep.

Item	HADG	LADG	*p*-Value
Dressing percentage/%	45.44 ± 1.52	46.27 ± 1.6	0.427
Carcass weight/kg	22.42 ± 2.51	19.46 ± 1.65	0.059
Eye muscle area/cm^2^	11.75 ± 2.69	11.29 ± 2.37	0.782
GR value/mm	13.68 ± 1.93	15.41 ± 1.61	0.162
Back fat thickness/mm	3.85 ± 0.82	3.01 ± 1.14	0.218
Head weight/kg	2.7 ± 0.26	2.5 ± 0.21	0.227
Proportion of head weight to live weight before slaughter/%	5.48 ± 0.3 ^b^	5.95 ± 0.34 ^a^	0.049
Hoof weight/kg	0.86 ± 0.08	0.81 ± 0.06	0.266
Proportion of hoof weight to live weight before slaughter/%	1.75 ± 0.11 ^b^	1.93 ± 0.12 ^a^	0.049
Tare weight/kg	4.65 ± 0.38	4.23 ± 0.57	0.206
Proportion of tare weight to live weight before slaughter/%	9.47 ± 0.77	10.03 ± 0.7	0.265
Tail fat weight/kg	1.73 ± 0.06	1.38 ± 0.48	0.141
Proportion of tail fat to live weight before slaughter/%	3.54 ± 0.31	3.24 ± 0.89	0.503

Note: Same row without letters indicates insignificant differences (*p* > 0.05), different lowercase letters indicate significant differences (*p* < 0.05).

### 3.2. Analysis of Blood Antioxidant Indices and Blood Biochemical Indices in High- and Low-ADG Tan Sheep

As shown in Table 4, the plasma T-AOC and SOD levels in the HADG group of *Tan* sheep were significantly higher than those in the LADG group (*p* < 0.05), while the MDA level was significantly lower than that in the LADG group (*p* < 0.05). There was no difference in GSH-Px between the two groups (*p* > 0.05). As indicated in Table 5, the plasma GLB, GLU, and TG levels in the HADG group of *Tan* sheep were significantly lower than those in the LADG group (*p* < 0.05), while other blood biochemical indices showed no significant differences (*p* > 0.05).

### 3.3. Analysis of Rumen Microbial Differences in High- and Low-ADG Tan Sheep

Paired-end sequencing was performed on the microbial DNA fragments from the rumen samples of *Tan* sheep. After screening and filtering, a total of 1,856,680 raw reads were generated from the rumen, with an average of 64,023 raw reads per sample. The number of sequences detected in the samples and their quality were satisfactory, allowing for subsequent analysis. In the rumen microbiota of the HADG and LADG groups, the HADG group obtained 796 Operational Taxonomic Units (OTUs), while the LADG group obtained 828 OTUs, with 652 OTUs shared between the two groups (Figure 1a). Alpha diversity index analysis provided information on species richness and diversity within the community. Shannon indicated that the total number of classifications in the samples and their proportions, as well as the Shannon, Simpson, Ace, and Chao 1 indices among microbial groups, showed no significant differences (*p* > 0.05) (Table 6). Analysis of the beta diversity of microbial communities revealed not certain but not significant separation between the rumen samples of the HADG and LADG groups of *Tan* sheep, indicating minor differences in microbial structure composition (Figure 1b,c). Further comparison of the rumen microbial classifications between the HADG and LADG groups showed similarities in their microbial structures. In the study at the phylum level, the dominant phyla in the rumen with a proportion greater than 5% were Bacteroidota and Firmicutes, with Bacteroidota exhibiting the highest relative abundance exceeding 50%, thus holding an absolute dominance (Figure 1d). At the family level, the analysis revealed that the dominant families in the rumen with a proportion greater than 5% were Prevotellaceae, Lachnospiraceae, and Erysipelotrichaceae, among which Prevotellaceae showed the highest relative abundance around 50%, demonstrating an absolute dominance (Figure 1e). In the study at the genus level, the dominant genera in the rumen microbiota with an abundance greater than 5% were Prevotella, norank_f__Prevotellaceae, and UCG-002, among which Prevotella had the highest relative abundance, exceeding 30%, and held an absolute dominance (Figure 1f). Differential genus analysis was performed on the top 15 microorganisms with relative abundance in the rumen of the HADG and LADG groups, identifying four significantly different genera: norank_f__Bacteroidales_UCG-001, UCG-002, norank_f__norank_o__Rhodospirillales, and Blautia (*p* < 0.05) (Figure 1g).

### 3.4. Metabolomic Analysis of Rumen Fluid in High- and Low-ADG Tan Sheep

PCA results showed that the HADG group and the LADG group exhibited good clustering with intersections, accompanied by a clear separation trend, indicating the presence of different metabolites (Figure 2a). OPLS-DA analysis revealed significant differences between the HADG group and the LADG group, with each group located in distinct regions. Samples within each group were tightly clustered, and the inter-group distance was significant, demonstrating robust stability and predictive capability, effectively distinguishing the rumen metabolite composition between the HADG group and the LADG group in *Tan* sheep (Figure 2b). Based on the criteria of FC ≥ 1.5 or FC ≤ 0.66, VIP > 1, and *p* < 0.05, differential metabolites were screened in the rumen of *Tan* sheep between the HADG and LADG groups, revealing a total of 265 differential metabolites. Compared with the HADG group, the LADG group showed 64 downregulated and 201 upregulated differential metabolites in the rumen fluid (Figure 2c). ROC curve analysis identified 15 significantly different metabolites, including L-proline, β-alanine, and carnitine, among others (Figure 2d, Table 7). Kyoto Encyclopedia of Genes and Genomes (KEGG) analysis results indicated that the differential metabolites in the rumen fluid of *Tan* sheep were significantly enriched in metabolic pathways such as sulfur metabolism, tyrosine metabolism, β-alanine metabolism, and thiamine metabolism (Figure 2e).

### 3.5. Correlation Analysis of Rumen Microbiota and Differential Metabolites in Rumen Fluid Between High- and Low-ADG Tan Sheep

The analysis of the correlation between differential microbial genera and differential metabolites in the rumen of HADG and LADG groups of *Tan* sheep showed that UCG-002 and norank_f__Bacteroidales_UCG-001 were significantly negatively correlated with Succinic acid (*p* < 0.01), and significantly negatively correlated with 1-Aminocyclopropanccarboxylic acid, Pyridoxal phosphate, and 2-Hydroxyglutarate (*p* < 0.05). Beta-Alanine, Ureidoacrylic acid, L-Proline, and 2′-Deoxyguanosine were significantly positively correlated with norank_f__Bacteroidales_UCG-001 (*p* < 0.01), and significantly positively correlated with UCG-002 (*p* < 0.05) (Figure 3).

## 4. Discussion

Integrated analysis of growth performance, serum biochemistry, ruminal microbiome, and metabolome provided a systematic understanding of the multi-tiered physiological profile associated with differences in average daily gain (ADG) in *Tan* sheep. A core finding of our study is that high-ADG individuals harbored a more efficient ruminal ecosystem, characterized by a specific microbial-metabolite interaction network. This network likely contributes to superior growth performance by collectively optimizing host energy metabolism efficiency and antioxidant status.

### 4.1. Growth Performance, Carcass Traits, and Systemic Physiological Status

Consistent with expectations, *Tan* sheep in the HADG group exhibited significantly higher average daily gain (ADG), average daily feed intake (ADFI), and improved feed conversion ratio (FCR), aligning with findings reported in other species [14,15]. Although pre-slaughter live weight differed significantly between the groups, no significant differences were observed in carcass weight or dressing percentage. This discrepancy may be attributed to variations in body frame size and the partitioning of body composition [16,17,18,19]. Visceral organ weight is generally indicative of metabolic intensity and physiological function. HADG individuals often possess heavier internal organs, implying heightened metabolic activity [20]. Our results revealed no significant differences in liver and lung weights between HADG and LADG *Tan* sheep. However, kidney weight was markedly and significantly greater in the HADG group compared to the LADG group, a finding consistent with prior research outcomes [21]. Oxidative stress leads to the overproduction of H2O2. T-AOC and SOD are crucial biomarkers for evaluating the systemic antioxidant status. MDA being an end-product of lipid peroxidation, serves as an indicator of oxidative damage to lipids. GSH-Px safeguards the structural and functional integrity of cell membranes by eliminating lipid hydroperoxides and mitigating membrane lipid oxidation [22,23]. At the systemic physiological level, HADG individuals demonstrated a stronger overall antioxidant capacity. This was evidenced by significantly higher plasma levels of T-AOC and SOD activity, coupled with lower concentrations of the lipid peroxidation product MDA [24,25,26]. Collectively, these findings suggest that HADG Tan sheep possess a enhanced overall antioxidant capacity relative to the LADG group. Plasma parameters including GLB, ALB, GLU, TG and TC are indicative of the metabolic and nutritional condition of the organism [27]. Research has demonstrated a significant correlation between TP and ALB levels and the efficiency of protein metabolism [28]. Yu et al. [29] documented that an increase in ADG in broilers was associated with raised levels of TP or TC. Concurrently, lower plasma concentrations of glucose (GLU) and triglycerides (TG) in the HADG group suggest a distinct pattern of energy substrate utilization [30,31]. Notably, despite these marked differences in redox status, circulating levels of inflammatory cytokines (IL-2, IL-6, IL-10) and immunoglobulins (IGA, IgG, IgM) remained comparable between groups. This indicates that the observed growth variation in our model likely stems primarily from intrinsic differences in the regulation of metabolic efficiency and redox homeostasis, rather than being driven by systemic inflammation.

### 4.2. Association Between Rumen Microbial Community Structure and ADG

The ruminal microbiota plays a pivotal role in nutrient digestion and host energy harvest [32]. Our analysis revealed that while no significant differences in alpha- and beta-diversity of the ruminal microbiota were observed between the HADG and LADG groups, significant shifts occurred in the abundance of specific taxa [33,34]. At the phylum level, *Bacteroidota* and *Firmicutes* predominated, which aligns with the common features of the gastrointestinal microbiome in ruminants [35,36]. Bacteroidota plays a crucial role in the breakdown of complex carbohydrates and the maintenance of gut homeostasis, enhancing carbohydrate utilization. Firmicutes, rich in metabolic enzymes, are the main microbiota involved in the degradation of fibrous materials such as cellulose and lignin [37]. Notably, variations in the relative abundance ratio of Firmicutes to Bacteroidota (F/B ratio) are often linked to the host’s energy acquisition efficiency. Myer et al. [38] found in cattle that an increased F/B ratio is associated with rumen energy storage and fat deposition in ruminants. At the family level, Prevotellaceae, Lachnospiraceae, and Erysipelotrichaceae are the dominant families. This finding is consistent with Ren et al. [39], who identified Prevotellaceae as the dominant family in the mouse gut, and Karstrup et al. [40], who reported a positive correlation between Lachnospiraceae abundance and ADG in beef cattle, as well as Stanley et al. [41], who found Erysipelotrichaceae as the dominant family in the chicken gut. Prevotellaceae is a core family in the rumen, playing an important role in the breakdown of fibrous foods. Some of its members contribute to promoting growth and enhancing feed efficiency [39,40,41]. At the genus level, *Prevotella* was the predominant genus, whose functional repertoire includes the degradation of polysaccharides and proteins [42,43,44]. Several differentially abundant genera were significantly associated with the HADG phenotype. Notably, the genus *Blautia* was enriched in the HADG group, a finding consistent with observations in Holstein cattle [30]. *Blautia* is reported to possess anti-inflammatory properties and may contribute to gastrointestinal health [45]. Furthermore, an unclassified genus belonging to the order norank_f__norank_o__Rhodospirillales was significantly increased in the HADG group. Despite its low relative abundance, its putative proteolytic activity could potentially contribute to nitrogen metabolism [46,47].

### 4.3. Key Metabolic Pathways Revealed by the Rumen Metabolomic Profiles

Metabolomic analysis identified a total of 265 differential metabolites, among which 201 were relatively upregulated in the LADG group. KEGG enrichment analysis indicated that tyrosine metabolism, β-alanine metabolism, and thiamine (vitamin B1) metabolism were the most significantly perturbed pathways. Key differential metabolites, such as succinic acid, were downregulated in the LADG group. As an essential intermediate of the tricarboxylic acid (TCA) cycle, reduced levels of succinic acid may suggest a lower metabolic flux and compromised energy production in LADG individuals [48,49]. Meanwhile, 2-hydroxyglutarate was elevated in the LADG group. This metabolite is known to act as a competitive inhibitor of succinate dehydrogenase in the TCA cycle, and its accumulation may further impair the efficiency of energy generation [48]. In contrast, pyridoxal phosphate (the active form of vitamin B6) was present at higher levels in the HADG group. Vitamin B6 functions as a coenzyme in numerous enzymatic reactions, particularly in amino acid metabolism; therefore, its increased abundance may be associated with more efficient nitrogen metabolism and protein utilization [50,51]. In addition, amino acids and related metabolites—including L-tyrosine, L-proline, and β-alanine—showed significant alterations in the LADG group, reflecting potential differences between the two groups in nitrogen metabolism and amino acid conversion efficiency [32,52,53,54,55].

### 4.4. Microbe–Metabolite Correlation Network: Establishing Functional Linkages

The rumen microbiome plays a crucial role in host health by providing metabolic products, maintaining metabolic functions, supporting the immune system, and defending against pathogens [56,57]. In goats, the abundance of Ruminococcus in the rumen is positively correlated with ADG [57]. The order Oscillospirales mainly produces butyrate and is positively associated with weight gain [58,59]. To elucidate the functional outputs associated with microbial alterations observed in this study, we conducted correlation analyses between key differential genera and representative differential metabolites. The results revealed a robust correlation network linking rumen microbiota to specific metabolic signatures. The differential genera UCG-002 and norank_f__Bacteroidales_UCG-001 exhibited a highly significant negative correlation with succinic acid. Beyond serving as an essential intermediate in energy metabolism, succinic acid has also been reported to participate in intestinal immune regulation [60,61]. Compared to HADG *Tan* sheep, succinic acid levels were significantly lower in the rumen fluid of LADG *Tan* sheep, while the abundance of UCG-002 and norank_f__Bacteroidales_UCG-001 in the rumen microbiota was significantly higher. These two genera may be responsible for the downregulation of succinic acid. These two genera were negatively correlated with pyridoxal phosphate and 2-hydroxyglutarate as well. In contrast, they showed a significant positive correlation with β-alanine, a metabolite upregulated in the LADG group and associated with muscle antioxidant capacity and nitrogen utilization [62]. Such strong correlations suggest that the distinct microbial community structures in the rumen of HADG and LADG *Tan* sheep may be directly linked to functionally divergent metabolic profiles. These functional linkages may influence host energy utilization and redox homeostasis, thereby contributing to the observed differences in ADG.

The novelty of this study lies in the integration of multi-omics datasets in *Tan* sheep for the first time, enabling a systematic depiction of the rumen ecosystem associated with divergent ADG phenotypes. Moreover, we established statistical associations between specific microbial taxa and functional metabolites, providing potential biomarkers and theoretical foundations for future nutritional interventions or microbiome-targeted strategies aimed at improving growth efficiency in *Tan* sheep. Future studies could incorporate functional validation approaches—such as in vitro fermentation assays or metagenomic/transcriptomic analyses—to substantiate the causal relationships implied by these correlations and to explore precision nutritional strategies based on key metabolites or microbial taxa.

## 5. Conclusions

This study demonstrates that differences in ADG in *Tan* sheep are closely associated with coordinated shifts in rumen microbial composition, metabolic pathways, and systemic antioxidant capacity. HADG individuals exhibited a more efficient rumen ecosystem, characterized by beneficial microbial taxa, enhanced energy metabolism, and stronger redox homeostasis. The identified microbe–metabolite associations—particularly those involving UCG-002, norank_f__Bacteroidales_UCG-001, and key metabolites such as succinic acid and pyridoxal phosphate—provide potential biomarkers linked to growth performance.

## Figures and Tables

**Figure 1 microorganisms-13-02882-f001:**
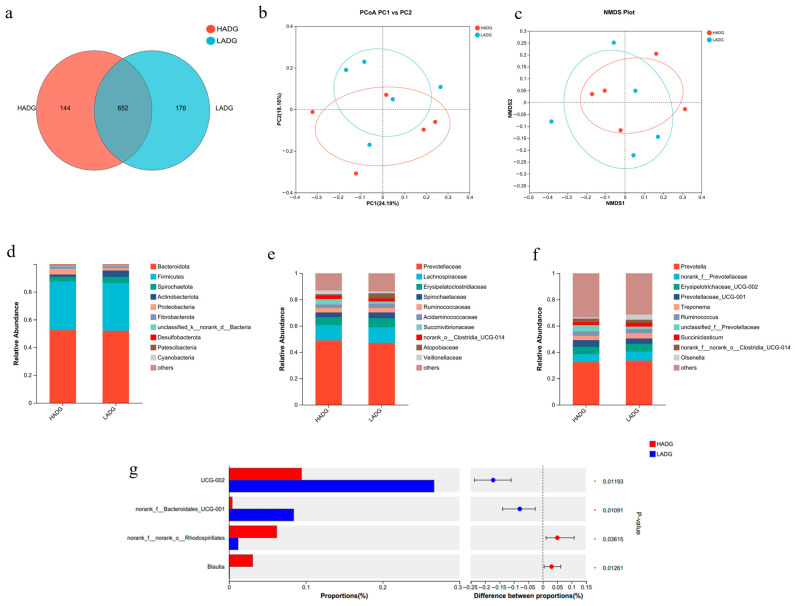
(**a**) LADG vs. HADG group OTUs Venn analysis; (**b**,**c**) LADG vs. HADG group principal co-ordinates analysis (PCoA) and Non-metric multidimensional scaling (NMDS) plots; (**d**–**f**) LADG vs. HADG group relative abundance at phylum, family, and genus levels; (**g**) LADG vs. HADG group plot of differential genera among microbes.

**Figure 2 microorganisms-13-02882-f002:**
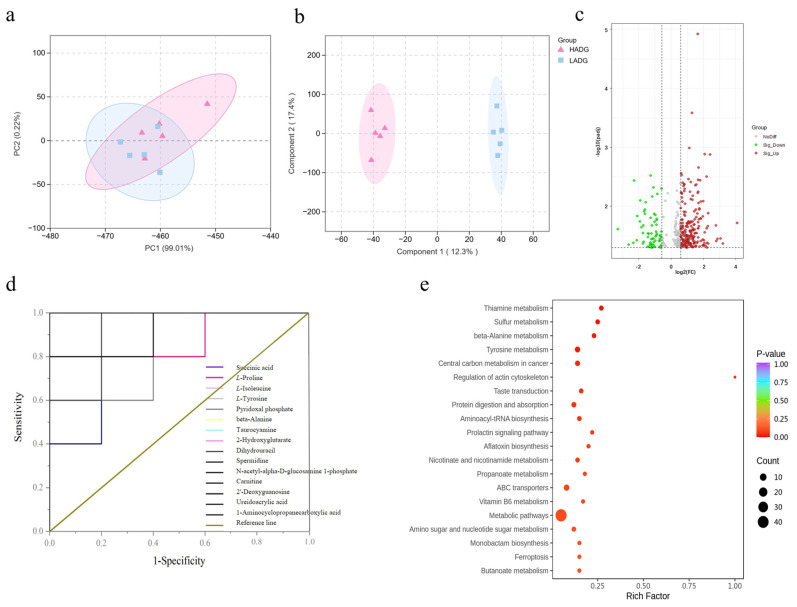
(**a**,**b**) LADG vs. HADG group PCA (Shows an overall separation in the global rumen metabolome between HADG and LADG groups, indicating fundamental phenotypic differences) and OPLS-DA plots (Statistically confirms the group separation and identifies the specific metabolites (high VIP values) most responsible for discriminating the two phenotypes); (**c**) LADG vs. HADG group volcano plot of differential metabolites (Details the specific dysregulation, quantifying which metabolites are significantly upregulated or downregulated in LADG sheep, indicating a broad shift in rumen metabolic output); (**d**) LADG vs. HADG group ROC curve of differential metabolites (Validate the diagnostic power of key differential metabolites, transforming them from statistical hits into potential functional biomarkers for ADG status); (**e**) LADG vs. HADG group metabolic pathways of differential metabolites (Provides the mechanistic link, showing that these key metabolites are enriched in specific pathways relevant to energy metabolism and antioxidant capacity—directly supporting the proposed biological basis for growth efficiency differences).

**Figure 3 microorganisms-13-02882-f003:**
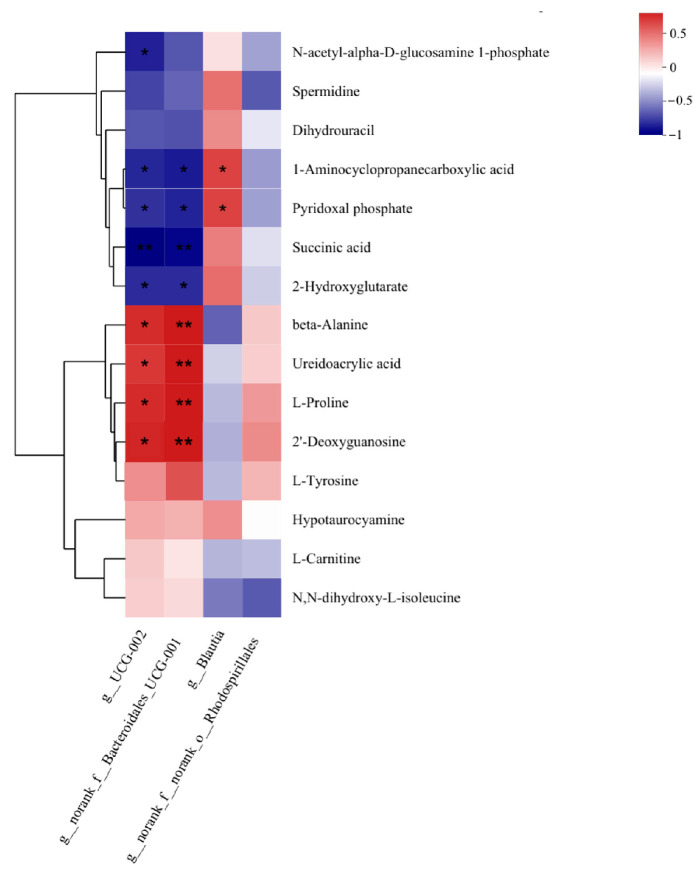
LADG vs. HADG group correlation between rumen microorganisms and rumen metabolites. * indicates significant correlation (*p* < 0.05); ** indicates extremely significant correlation (*p* < 0.01).

**Table 1 microorganisms-13-02882-t001:** Composition and nutrient levels of the basal diets.

Items	Content%
Ingredients	
Corn straw	20.00
Corn	32.00
Molasses	4.00
Soybean meal	6.00
Cotton meal	8.00
Corn bran	15.30
Corn germ meal	11.00
Limestone	1.20
NaCl	0.50
Premix ^1^	2.00
Total	100.00
Nutrient levels ^2^	
ME/(MJ/kg)	9.93
CP	13.74
EE	2.74
NDF	35.99
ADF	20.34
Ca	0.71
TP	0.31

^1^ The premix provides per kilogram of diet: 250,000 IU vitamin A, 375 IU vitamin E, 100,000 IU vitamin D, 850 mg iron, 800 mg copper, 750 mg zinc, 750 mg manganese, 25 mg selenium, 50 mg iodine, and 10 mg cobalt. ^2^ The metabolizable energy is calculated according to NY/T 816-2004 [11], and the rest are measured values. ME was calculated according to NY/T 816-2004, while the others were measured values.

**Table 4 microorganisms-13-02882-t004:** Analysis of blood antioxidant indices in high- and low-ADG *Tan* sheep.

Item	HADG	LADG	*p*-Value
T-AOC (μg/mL)	661.95 ± 122.01 ^a^	402.44 ± 149.79 ^b^	0.027
SOD (U/mL)	161.6 ± 19.95 ^a^	117.5 ± 31 ^b^	0.044
GSH-Px (μmol/L)	17.7 ± 3.04	13.44 ± 5.03	0.183
MDA (nmol/L)	6.13 ± 1.77 ^b^	10.34 ± 2.25 ^a^	0.019

Note: Same row without letters indicates insignificant differences (*p* > 0.05), different lowercase letters indicate significant differences (*p* < 0.05).

**Table 5 microorganisms-13-02882-t005:** Analysis of blood biochemical indices in high- and low-ADG *Tan* sheep.

Item	HADG	LADG	*p*-Value
IgA (μg/mL)	236.08 ± 40.28	232.08 ± 46.21	0.896
IgG (μg/mL)	483.23 ± 160.24	568.02 ± 130.04	0.408
IgM (μg/mL)	1667.18 ± 270.69	1621.78 ± 424.85	0.859
IL-2 (pg/mL)	981.55 ± 116.88	1353.8 ± 348.37	0.083
IL-6 (pg/mL)	118.9 ± 50.46	190.4 ± 51.84	0.076
IL-10 (pg/mL)	64.68 ± 5.71	45.74 ± 16.9	0.072
TP (μg/mL)	0.716 ± 0.050	0.723 ± 0.027	0.801
ALB (g/L)	64.86 ± 18.88	97.85 ± 24.36	0.062
GLB (g/L)	31.5 ± 7.25 ^b^	52 ± 10.22 ^a^	0.012
GLU (mmol/L)	4.85 ± 1.27 ^b^	8.03 ± 1.54 ^a^	0.013
TG (mmol/L)	5.78 ± 1.03 ^b^	8.6 ± 2.02 ^a^	0.040
TC (mmol/L)	7.28 ± 1.82	10.34 ± 2.27	0.065

Note: Same row without letters indicates insignificant differences (*p* > 0.05), different lowercase letters indicate significant differences (*p* < 0.05).

**Table 6 microorganisms-13-02882-t006:** Alpha diversity index table.

Item		HADG	LADG	*p*-Value
Rumen	Shannon	4.05 ± 0.16	4 ± 0.29	0.578
	Simpson	0.04 ± 0.005	0.05 ± 0.02	0.221
	Ace	491.12 ± 44.69	511.88 ± 86.3	0.646
	Chao 1	485.9 ± 47.33	502.59 ± 74.78	0.684

**Table 7 microorganisms-13-02882-t007:** Key differential metabolites in rumen of *Tan* sheep in LADG vs. HADG groups.

Metabolites	VIP	*p*-Value	FC
L-Proline	2.62	<0.001	2.43
beta-Alanine	2.19	0.046	2.31
Carnitine	2.25	0.003	2.11
Taurocyamine	1.90	0.022	2.04
2′-Deoxyguanosine	1.88	0.050	1.91
Ureidoacrylic acid	2.45	0.006	1.75
L-Isoleucine	1.74	0.041	1.61
L-Tyrosine	1.96	0.025	1.51
Succinic acid	2.04	0.033	0.66
1-Aminocyclopropanecarboxylic acid	1.82	0.031	0.63
Dihydrouracil	1.90	0.050	0.63
2-Hydroxyglutarate	2.53	0.007	0.44
Spermidine	1.80	0.045	0.43
Pyridoxal phosphate	2.34	0.003	0.42
N-acetyl-alpha-D-glucosamine 1-phosphate	2.04	0.045	0.40

## Data Availability

The original contributions presented in this study are included in the article/Appendix A. Further inquiries can be directed to the corresponding author.

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
