# Peer review of "Integrative Multi-Omics Analysis of the Rumen in Tan Sheep with Contrasting Average Daily Gain"

_microorganisms, 2025, doi:10.3390/microorganisms13122882_

Round 1
Reviewer 1 Report
Comments and Suggestions for Authors
The manuscript titled “Analysis and Correlation of Growth Performance, Blood Biochemical Features, Rumen Metabolomics, and Rumen Microbiome Differences between High and Low ADG Tan Sheep” presents an integrative analysis combining growth traits, blood biochemistry, oxidative stress and immune markers, rumen microbiota (16S rRNA), and untargeted metabolomics to identify factors associated with variations in average daily gain (ADG).
The study has potential scientific value; however, substantial revision is required before it can be considered for publication.
- Title
- The title is excessively long and reads like a complete summary.
- Suggestion: Condense it to focus on the core variables.
- Summary
- No summary is provided. Many journals require a short bullet-point summary or highlights.
- Authors should add a concise summary highlighting the research question, methods, key findings, and significance.
- Abstract
- Contains the essential components; however, the novelty of the study needs to be clearly stated.
- The abstract should avoid excessive background detail and emphasize the major findings and their implications.
- Introduction
- The literature review reads like a thesis: too long, descriptive, and unfocused.
- The introduction should:
- Clearly define the problem statement,
- Identify knowledge gaps,
- State why ADG variation in Tan sheep matters,
- Provide a strong and specific novelty statement.
- Currently, the novelty is unclear.
- Materials and Methods
- Slaughtering of animals
- Line 131–132: “After a 12-hour fast, the lambs were slaughtered…”
- Rationale for slaughter must be justified.
- Authors should explain why rumen sampling could not be done via rumen fistula or post-mortem collection at abattoir.
- Ethical considerations must be stated.
- Ethical approval
- Ethical approval for animal handling, slaughter, and sample collection is missing.
- This is mandatory for publication.
- Experimental design
- Overall acceptable, but missing clarity in some sections.
- Slaughter performance measurement
- Methods used for recording “differences in slaughter performance” are not specified.
- Authors must detail the parameters measured (carcass weight, dressing %, organ weights), instruments used, and the protocol followed.
- Oxidative stress parameters
- The reason for selecting oxidative stress markers (MDA, SOD, GSH-Px, etc.) is unclear.
- Authors should justify:
- Was oxidative stress expected in high or low ADG groups?
- How does oxidative stress relate to growth differences?
- Interpretation of oxidative stress
- If the study did not induce stress and animals were not under thermal or nutritional stress:
- Why were MDA levels elevated?
- This needs explanation or re-evaluation.
- Immunity parameters
- The rationale for measuring ILs and other immune markers is unclear.
- If immune status is not part of the hypothesis, these data appear out of place.
- Authors must justify their inclusion or remove irrelevant parameters.
- Choice of blood parameters
- Instead of oxidative and immune indicators, general metabolic markers (energy metabolism, protein metabolism, hormones, growth regulators) would have been more relevant to explain ADG variation.
- Results
- Figures and tables
- Quality needs substantial improvement:
- Increase resolution,
- Ensure consistent formatting,
- Improve color contrast,
- Use readable axis labels.
- Abbreviations
- Many abbreviations are not explained either in figure legends or at first appearance.
- All abbreviations must be defined.
- Clarity of Figures 2c, 2d, 2e
- These panels are unclear and do not convey a meaningful message.
- The authors must explain:
- What biological significance these graphs show,
- How they support the hypothesis regarding ADG differences.
- Linking results to hypothesis
- Results lack a clear narrative connecting microbial/metabolite changes to growth performance outcomes.
- Discussion
- The discussion is written in a thesis-like descriptive style, not a concise, hypothesis-driven discussion.
- Major issues:
- Immune markers and ILs that showed trends are not discussed.
- Authors have discussed general microbiota abundance without relating it to ADG variation.
- The discussion should focus on:
- Mechanistic links between microbiota–metabolites–growth,
- Biological relevance of identified biomarkers,
- Integration of findings rather than stand-alone descriptions from previous literature.
- Conclusion
- Too long and repetitive.
- Should be reduced to 4–6 lines summarizing key findings and future implications.
The manuscript required professional rewriting.
Reviewer 2 Report
Comments and Suggestions for Authors
Manuscript details:
Journal: Microorganisms
Manuscript ID: microorganisms-4020132
Type of manuscript: Article
Title: Analysis and Correlation of the Growth Performance, Blood biochemical features, Rumen metabolomics, and Rumen microbiome differences between High and Low ADG Tan sheep
Authors: Hao Zheng , Xiaohong Han , Wenjuan Shen , Xinrui Zhang , An Shi , Tonggao Liu , Chong Yang , Jinzhong Tao *
The authors aim of this research is to compared Tan sheep with high (HADG) and low (LADG) average daily gain using metabolomic and 16S rRNA sequencing approaches. They found that significant differences in rumen microbiota and metabolites were observed between HADG and LADG groups, providing a basis for elucidating the molecular mechanisms underlying growth traits and suggesting potential strategies for reducing breeding costs through optimized feeding regimens.
While this work is valuable, several concerns were raised throughout the text and need to be clarified by the authors before further consideration.
Please refer to the comments in the PDF file and revise accordingly before further consideration.

Reviewer 3 Report
Comments and Suggestions for Authors
This article provides information on the growth performance, blood biochemical parameters, rumen metabolomics, and rumen microbiome differences between High and Low ADG Tan sheep. It is in general appropriately organized, carried out and written, however there are some points that should be corrected or clarified. Please check comments and corrections in the attached file.

Round 2
Reviewer 1 Report
Comments and Suggestions for Authors
The authors have improved the quality of the manuscript and have answered most of the queries.
Comments on the Quality of English LanguageIt has been substantially improved.
Reviewer 2 Report
Comments and Suggestions for Authors
All concerns were addressed, no further comment.